# Deletion of the *sodA* Gene Impaired the Pathogenicity of *Streptococcus suis* Serotype 2 to Mice by Inhibiting Caspase-1/GSDMD Pathway Activation in Macrophages

**DOI:** 10.3390/microorganisms13112566

**Published:** 2025-11-10

**Authors:** Yajuan Li, Qiuguo Fang, Peiran Feng, Yushu Li, Qinqin Sun, Yunfei Huang, Shun Li, Oladejo Ayodele Olaolu, Qiang Fu

**Affiliations:** 1School of Animal Science and Technology, Foshan University, Foshan 528225, China; liyajuan@fosu.edu.cn (Y.L.); fangqiuguo32@163.com (Q.F.); egg19961126@icloud.com (P.F.); 15797954937@163.com (Y.L.); sunqinqin@fosu.edu.cn (Q.S.); huangyunfei@fosu.edu.cn (Y.H.); 2Foshan University Veterinary Teaching Hospital, Foshan University, Foshan 528225, China; 3Kunyu Integrated Agricultural Research Insititute, Xinjiang Academy of Agricultural and Reclamation Science, Shihezi 848116, China; lishunlucky@163.com; 4Department of Animal Health Technology, Oyo State College of Agriculture and Technology, Igboora 201003, Nigeria; oladejoayodele85@gmail.com

**Keywords:** *Streptococcus suis* serotype 2, *sodA* gene, macrophage

## Abstract

*Streptococcus suis* serotype 2 (SS2) is a major zoonotic pathogen causing infectious disease in various species, whose pathogenesis is still not well understood. The *sodA* gene is an important virulence gene of SS2 involved in the host’s immune response against pathogens. This study aimed to explore the impact of *superoxide dismutase A* (*sodA*) gene deletion on the pathogenicity of SS2 to mice. In this study, mice were grouped as control, WT, and Δ*sodA*, which were intraperitoneally injected with PBS, wild-type strain HA9801, and Δ*sodA* strain, respectively. WT proved to be a more virulent strain to mice with higher bacterial loads and survival rates in mice than those for Δ*sodA*. Moreover, more-severe tissue damage was observed in the lungs, liver, spleen, and kidneys of mice injected with WT than with Δ*sodA*. Additionally, macrophages accumulate to defend against SS2, and the results indicated that *sodA* gene deficiency decreased macrophage recruitment. In in vitro studies, caspase-1 and gasdermin D (GSDMD) were activated in macrophages induced by SS2; however, the absence of the *sodA* gene significantly inhibited the expression of pro-caspase-1, caspase-1, and GSDMD-N. Moreover, deletion of the *sodA* gene also decreased Interleukin-1 beta (IL-1β) and Interleukin-18 (IL-18) release in macrophages induced by SS2. Taken together, the absence of the *sodA* gene alleviated the pathogenicity of SS2 as a result of decreased macrophage accumulation and breakage of the caspase-1/GSDMD pathway in macrophages.

## 1. Introduction

*Streptococcus suis* is the etiological agent responsible for various illnesses, including arthritis, endocarditis, meningitis, and septicemia, in many species including humans, pigs, and mice [1]. It is also recognized as a zoonotic pathogen implicated in human meningitis and severe toxic shock syndrome [2]. Among the identified serotypes of *Streptococcus suis*, serotype 2 (SS2) is the most virulent and prevalent [3]. Furthermore, SS2 is regarded as an emerging pathogen, posing a significant threat to the global swine industry.

Virulence factors serve as critical determinants through which pathogenic microorganisms engage with the host; consequently, it is essential to investigate the role of these factors during the course of infection. Efforts have been made to characterize virulence factors as part of the strategy to reveal the mechanisms of SS2 infections [4,5]. The identified virulence factors of SS2 to date encompass the antiphagocytic capsular polysaccharide, suilysin, cell wall-associated and extracellular proteins, fibronectin-binding proteins, and the serum opacity factor [6]. However, the pathogenesis of SS2 remains insufficiently elucidated, prompting further investigation into its virulence factors.

Macrophages are pivotal components of the innate immune system, playing a crucial role in host defense mechanisms during infections [7]. A notable feature of macrophages is their ability to respond to extracellular signals, enabling them to assume various functional roles in alignment with these stimuli [8]. Macrophages respond to invading microorganisms through a respiratory burst, resulting in the production of highly microbicidal reactive oxygen species (ROS), such as superoxide anions (O^2−^) and hydroxyl radicals (OH^−^) [9]. ROS induce deleterious alterations in DNA, RNA, proteins, and lipids [10]. Bacteria consistently encode antioxidant enzymes, such as superoxide dismutases (SODs) and catalase, to counteract reactive oxygen species (ROS) generated by the host, thereby promoting their survival [11]. The *sodA* gene, a critical gene encoding SODs, has been reported to not only scavenge ROS but to also influence the immune response of macrophages. Specifically, the *sodA* gene of SS2 has been shown to mitigate host autophagic responses to enhance bacterial survival [12]. However, the precise role of the *sodA* gene in modulating the immune response induced by SS2 remains unclear.

Pyroptosis plays a crucial role in the defense mechanisms of macrophages against pathogenic infections [13], initially characterized as a process dependent on the activation of caspase-1 and the involvement of gasdermin D (GSDMD). Pyroptosis augments immune defense mechanisms by disrupting the protective intracellular replication niches utilized by pathogens. While moderate pyroptosis serves as a host defense strategy against infectious agents, excessive pyroptosis can be linked to disproportionate inflammatory responses and significant tissue damage [14]. Macrophage pyroptosis has been evidenced to be involved in the host defense against SS2 [15,16]; however, only the *sly* gene of SS2 has been revealed to contribute to macrophage pyroptosis [16]. Therefore, it is important to investigate additional virulence genes of SS2 that are implicated in host macrophage pyroptosis. A previous study indicated that the *sodA* gene played an important role in the autophagy of macrophages induced by SS2, which inspired us to explore whether the *sodA* gene is involved in the pyroptosis of macrophages [17].

This study aimed to investigate the role of the *sodA* gene in the pathogenicity of SS2 to mice. A *sodA* gene deletion mutant (Δ*sodA*) and the wild-type strain HA9801 (WT) were used to compare the virulence of Δ*sodA* and WT in mice. The results indicated that the *sodA* gene was important to the pathogenicity of SS2 on account of accelerating macrophage accumulation and pyroptosis.

## 2. Materials and Methods

### 2.1. Mice Infection

Six- to eight-week-old male C57BL/6J mice were purchased from Guangdong Medical Laboratory Animal Center (Guangzhou China). The mice were bred in the animal facility of the Laboratory Animal Center (South China Agricultural University, Guangzhou, China). The mice were kept in plastic cages with standard bedding, fed standard rodent chow, and provided water ad libitum. Wild-type strain HA9801 is a clinical strain from the lung of a diseased pig. Unless otherwise indicated, SS2 strains were grown in Brain Heart Infusion (BHI, Oxoid, Basingstoke, UK) or LB agar Luria–Bertani (LB) broth at 37 °C. For bacterial counts, a single colony from TSA agar plates of WT and Δ*sodA* strains was transferred to 5 mL TSB broth and incubated at 37 °C for 12 h. Then, 250 μL of each culture was added to 25 mL fresh TSB and incubated at 37 °C and 220 rpm, and the manual counting method was used for bacterial counts. The Δ*sodA* was kindly donated by Dr. Fang [17]. To observe the survival rate of infected mice, mice were intraperitoneally infected with 500 μL 1×10^8^ CFU of WT or Δ*sodA* (*n* = 10). Mice intraperitoneally infected with 500 μL PBS were regarded as healthy controls (*n* = 10). The mice were observed for 96 h to record the survival rate.

To elucidate the role of *sodA* in cellular proliferation, a single colony from the TSA agar plates inoculated with WT strains and Δ*sodA* strains was transferred into 5 mL of TSB broth and incubated at 37 °C for 12 h. Each calibrated culture (250 μL) was inoculated into 25 mL of fresh TSB medium and incubated at 37 °C and 220 rpm. A volume of 100 μL was collected from each culture at 2 h intervals over a period of 14 h. The plate colony counting method was used for colony counting of WT and Δ*sodA* strains. CFU values at each time point were calculated as the mean from three independent experiments.

### 2.2. Bacterial Loads

The PLF suspension and blood were introduced into sterile TSA and incubated at 37 °C for 14 h. The manual counting method was employed to determine the number of single colonies exhibiting a morphology consistent with that of WT and Δ*sodA* colonies.

### 2.3. Tissue Histology Examination

A paraformaldehyde 4% fixative was used to fix the lung tissues, and paraffin wax was used to embed them. For histology, lung, liver, spleen, and kidney tissue sections (2.0 × 2.0 × 0.3 cm) were stained with hematoxylin and eosin (H&E).

### 2.4. Flow Cytometry and Western Blotting

For macrophage counting, the cells from PLF were stained for 30 min with Anti-F4/80 (1:80, BM8, eBioscience, San Diego, CA, USA) and Anti-CD11b (1:100, M1/70, eBioscience, San Diego, CA, USA) after being blocked with Anti-CD16/CD32 (1:62, 93, eBioscience, San Diego, CA, USA); the result was obtained with a CytoFLEX flow cytometer (Beckman Coulter, Brea, CA, USA). The software CytExpert 2.3 was used to analyze the data.

Total protein samples from the lung tissue were collected using the Thermo Scientific TPER Kit. These protein samples were subsequently separated by SDS-PAGE and transferred onto PVDF membranes. Following a blocking step with 5% milk, the membranes were incubated with a primary antibody, followed by incubation with an HRP-conjugated secondary antibody (ab288151, Abcam, Cambridge, UK). Detection of the results was performed using a BIO-RAD ChemiDoc^TM^ MP imaging system (Alfred Nobel Drive, Hercules, CA, USA). Band intensities were quantified using ImageJ 1.54F software, with the values presented below each band indicating the average ratio of the protein or modification relative to the loading control. The primary antibodies utilized in this study included those against GSDMD (ab209845, Abcam), IL-18 (ab207323, Abcam), IL-1β (ab283818, Abcam), caspase-1 (ab138483, Abcam), cleaved-caspase-1 (APG02632G, AdipoGen, San Marino, Switzerland), and β-actin (ab8226, Abcam).

### 2.5. Cytokine Assays

Cell supernatant samples were used for IL-1β and IL-18 assays. ELISA kits were used to detect these cytokines according to the manufacturer’s protocols. The ELISA kits were purchased from Shanghai Enzyme-linked Biotechnology Co. (Shanghai, China).

### 2.6. Statistical Analysis

All data are represented as the mean ± SD. The data was analyzed using GraphPad Prism (v8.0.1, GraphPad Software, San Diego, CA, USA); data from different groups were compared by one-way ANOVA, with *p* < 0.05 being considered statistically significant and *p* < 0.01 being highly significant.

## 3. Results

### 3.1. Deficiency of the sodA Gene Impaired the Pathogenicity of SS2 to Mice

As shown in Figure 1A, Δ*sodA* exhibited slower growth than the WT stains with *p* < 0.01 at hour 10 post-inoculation. The impact of *sodA* deletion on virulence was assessed in BALB/c mice. All mice inoculated with WT strains showed signs such as ruffled hair coat and slow response to stimuli and died within the first 36 h. However, most of the mice inoculated with Δ*sodA* did not show clinical symptoms and stayed alive, with only two deaths during the first 24 h (Figure 1B). For further study, the bacterial loads of WT and Δ*sodA* in PLF and blood samples collected from mice at 24 h after intraperitoneal injection were investigated. The Δ*sodA* loads were higher than those for WT both in blood and PLF, with *p* < 0.01 (Figure 1C,D). These results confirmed that deletion of the *sodA* gene significantly impaired the virulence of SS2 in mice. And the decreased pathogenicity of *sodA* may result from the lower bacterial loads in mice.

### 3.2. Deficiency of the sodA Gene Alleviated Tissue Damage in Mice from SS2

Further studies investigated the effect of *sodA* gene on tissue injury induced by SS2. As shown in Figure 2, HE staining showed that the lung, spleen, liver, and kidney tissues were evidently impaired by WT, and mice injected with WT showed more-severe tissue damage than the Δ*sodA* group. In detail, WT caused obvious thickening of alveolar walls, collapse of alveoli, and hemorrhage in lung tissue; Δ*sodA* showed lessened tissue damage, and WT resulted in more-severe hemorrhage and slightly enlarged white pulp with mild lymphocyte proliferation in the spleen compared with Δ*sodA*. Furthermore, obvious hepatocellular damage and hemorrhage were seen in livers from mice infected with Δ*sodA;* in the kidney tissue, more-severe tubular vacuolar degeneration and increased blood cells were observed from WT-infected mice compared with that in those infected with Δ*sodA*. Additionally, macrophages were recruited in the lung, spleen, liver, and kidney tissues by SS2, and Δ*sodA* caused less macrophage infiltration than WT (Figure 2). These results suggest that deletion of the *sodA* gene alleviated the tissue damage caused by SS2. And macrophages are recruited to defend against SS2, but *sodA* gene deficiency decreases macrophage infiltration.

### 3.3. Deficiency of the sodA Gene Decreased Bacterial Loads of SS2 in Mice

Tissue damage always results from bacterial colonization [18]; this study therefore explored the bacterial loads in tissues collected from mice injected with WT and Δ*sodA*. As shown in Figure 3, more WT colonized in liver, lung, spleen, and kidney tissues of mice than Δ*sodA* (*p* < 0.05). These findings suggested that deletion of the *sodA* gene decreased the bacterial loads of SS2 in mice.

### 3.4. Deficiency of the sodA Gene Decreased Macrophage Infiltration in Mice Injected with SS2

Macrophages are crucial immune cells, and they cause obvious infiltration in tissues; the present study further detected the recruitment of macrophages in PLF samples from mice challenged with WT and Δ*sodA.* As shown in Figure 4A–C, we compared macrophage populations in PLF (CD11b^+^F4/80^+^) collected from mice injected with WT and Δ*sodA*. WT induced more macrophages in PLF than Δ*sodA* (Figure 4D). These data reveal that deletion of the *sodA* gene decreased the macrophages population in PLF induced by SS2.

### 3.5. Deficiency of the sodA Gene Impaired the Resistance to Phagocytosis and Killing in Macrophages

In mice, macrophages are involved in defense against SS2, and our previous results evidenced that absence of the *sodA* gene impaired the recruitment of macrophages. The present study further examined the effect of *sodA* gene deletion on phagocytosis in a macrophage cell line J774A.1. As shown in Figure 5, the Δ*sodA* mutant was more readily phagocytosed than WT in 1 h at MOI = 10, 20, 40, 80 (*p* < 0.05) in macrophages. These data suggest that the *sodA* gene was involved in SS2 against phagocytosis of macrophages.

### 3.6. Deficiency of the sodA Gene Alleviated the Caspase-1/GSDMD Pathway in Macrophages Induced by SS2

It is generally reported that an inflammatory response mediated by caspase-1/GSDMD is involved in host defense against bacterial invasion [19]. Therefore, this study examined whether *sodA* gene deletion influenced the caspase-1/GSDMD pathway in macrophages activated by SS2. As shown in Figure 6, the protein expression levels of GSDMD, GSDMD-N, caspase-1, cleaved-caspase-1, IL-1β, and IL-18 in macrophages incubated with WT were markedly higher than those in PBS-treated macrophages (*p* < 0.05); compared with WT, Δ*sodA* induced a lower expression of the aforesaid proteins (*p* < 0.05). These results indicate that deletion of the *sodA* gene inhibited caspase-1/GSDMD pathway activation in macrophages induced by SS2.

### 3.7. Deficiency of the sodA Gene Reduced Inflammatory Cytokine Release in Macrophages Induced with SS2

Activation of the caspase-1/GSDMD pathway always results in inflammatory cytokine release [20]. As expected, WT induced higher IL-1β and IL-18 in the supernatant than Δ*sodA* (*p* < 0.05) (Figure 7A,B). Additionally, WT induced higher LDH release in supernatants than Δ*sodA* (Figure 7C). Thus, *sodA* gene deficiency suppressed the secretion of inflammatory cytokines in macrophages induced by SS2.

## 4. Discussion

SS2 is acknowledged as the most virulent and frequently isolated serotype among all identified serotypes of *Streptococcus suis* in both animal and human populations [15]. Most existing studies of SS2 universally focus on the identification of virulence factors [21,22,23]. As virulence factors are important targets that mediate the interaction between microorganism and host [24], the impact of virulence factors on the host response to SS2 needs further studies. The *sodA* gene of SS2 was reported as an important virulence gene to trigger host immune response [25], and its specific role needs exploring. This study demonstrated that the absence of the *sodA* gene significantly attenuated the pathogenicity of SS2 in murine models, underscoring the critical role of the *sodA* gene in macrophage-mediated immune responses.

In this study, SS2 exhibited significant pathogenicity in mice, and the deletion of the *sodA* gene reduced the virulence of SS2 in these animals (Figure 1B). These findings demonstrate that the sodA gene plays a crucial role in SS2 infection, consistent with the previous research [17]. Bacterial colonization in a host is necessary for infection. We found that Δ*sodA* showed lower CFUs both in PLF and blood samples than WT (Figure 1D,E). SOD is an important virulence factor of various bacteria encoded by the *sodA* gene, which is involved in bacterial survival in the host [26,27,28]. Therefore, we supposed that deletion of the *sodA* gene impairs the survival of SS2 in mice. In addition, obvious tissue damage was observed in the lung, liver, spleen and kidney tissues of mice infected with SS2; distinguishingly, the lung exhibited the most severe signs (Figure 2). Consistent with the results in Figure 1, the *sodA* gene showed alleviated tissue injury and lower bacterial loads in tissue compared with SS2 (Figure 3). Taken together, we demonstrated that deletion of the *sodA* gene impaired the pathogenicity of SS2 because of the lower bacterial survival.

Macrophages play crucial roles in the phagocytic uptake of the majority of invading bacteria and utilize a variety of bacterial killing mechanisms to efficiently eliminate diverse bacterial pathogens [29]. In this study, macrophages were identified as playing a role in the defense of mice against SS2 (Figure 2 and Figure 4). This finding is consistent with previous research indicating that macrophages are recruited to the brain in response to SS2 [30], suggesting their involvement in the defensive mechanisms against SS2. Moreover, we initially found that WT recruited more macrophages than Δ*sodA* (Figure 4). While macrophages are the most plastic cells acting as the main and primary barrier in various tissues of the body against foreign invaders [31], massive stimulation of macrophages can lead to fatal septic shock syndrome and multiple organ failure [32,33,34]. For further study, we compared the phagocytosis of SS2 and Δ*sodA* in macrophages. The Δ*sodA* mutant was more-readily phagocytosed than SS2. Similarly, a prior study demonstrated that SS2 with a deletion of the sodA gene exhibited increased susceptibility to macrophage-mediated killing [35]. These results directly evidenced that the *sodA* gene is a crucial virulence gene of SS2 pathogenicity to mice, and the specific mechanism needs further study.

Pyroptosis, which was initially defined as caspase-1-dependent necrotic death [36], is morphologically and mechanistically distinct from other forms of cell death. Pyroptosis is characterized by rapid plasma membrane rupture and the release of proinflammatory intracellular contents [37,38]. Following cleavage, the N-terminal domain of GSDMD targets and enters cellular membranes to form large oligomeric membrane pores, disrupting ion homeostasis and ultimately causing cell death [39]. In the present study, caspase-1 and GSDMD in macrophages were activated by SS2; however, deletion of the *sodA* gene markedly inhibited caspase-1 and GSDMD activation (Figure 6). Thus, we proposed that the *sodA* gene was employed to promote macrophage pyroptosis induced by SS2. Accordingly, *sodA* gene deficiency of SS2 showed impaired resistance to phagocytosis by alleviated macrophage pyroptosis. Caspase-1/GSDMD pathway activation always leads to an inflammatory cascade with the release of cytokines [40]. The results in this study revealed that WT induced more cytokine release than Δ*sodA* in macrophages (Figure 7). It is reported that the suppression and failure of immune function in bacteria-infected animals results from macrophage pyroptosis [41]. Therefore, we demonstrated that Δ*sodA* inhibited macrophage pyroptosis for survival mediated by the *sodA* gene.

However, the absence of the CΔ*sodA* (complemented strain of Δ*sodA*) in this study to validate the findings is an apparent limitation in this study. Further research is necessary to achieve robust validation using CΔ*sodA*, which would more rigorously substantiate the findings of this study.

## 5. Conclusions

Taken together, we found deletion of the *sodA* gene impaired the pathogenicity of SS2 to mice, and *sodA* gene deficiency decreased macrophage accumulation and promoted phagocytosis of SS2 by macrophages. Additionally, the absence of the *sodA* gene alleviated macrophage pyroptosis induced by SS2. In conclusion, this study provides a significant insight into the role of the *sodA* gene in SS2 infection.

## Figures and Tables

**Figure 1 microorganisms-13-02566-f001:**
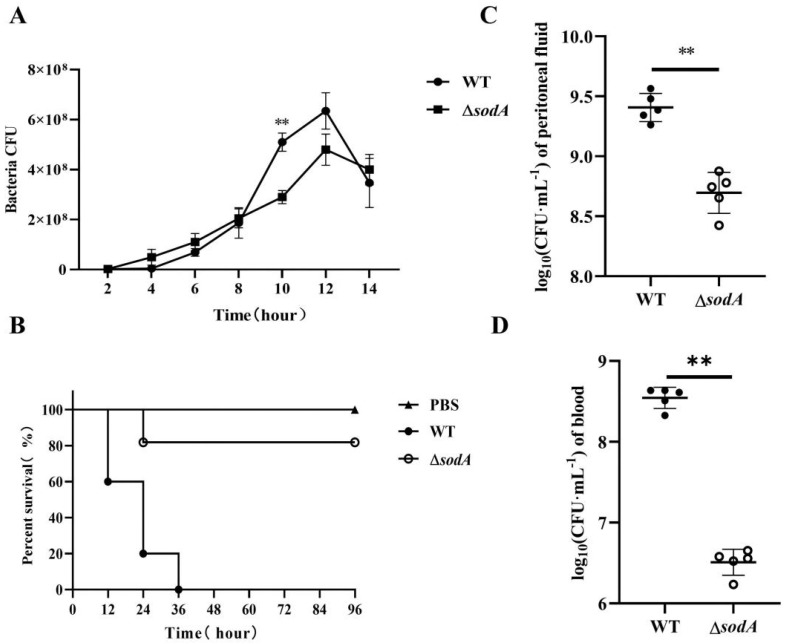
Effect of *sodA* gene deletion on the SS2 pathogenicity to mice. (**A**) Growth characteristics of WT strains and Δ*sodA* strains; (**B**) survival of the mice challenged with WT and Δ*sodA* (*n* = 10); survival data were analyzed using the log-rank test; (**C**,**D**) bacterial loads of WT and Δ*sodA* in peritoneal fluid and blood samples from mice inoculated with WT and Δ*sodA* at 24 h, *n* = 6. ** *p* < 0.01; one-way ANOVA with Tukey’s post hoc test was used; data are shown as mean ± SD.

**Figure 2 microorganisms-13-02566-f002:**
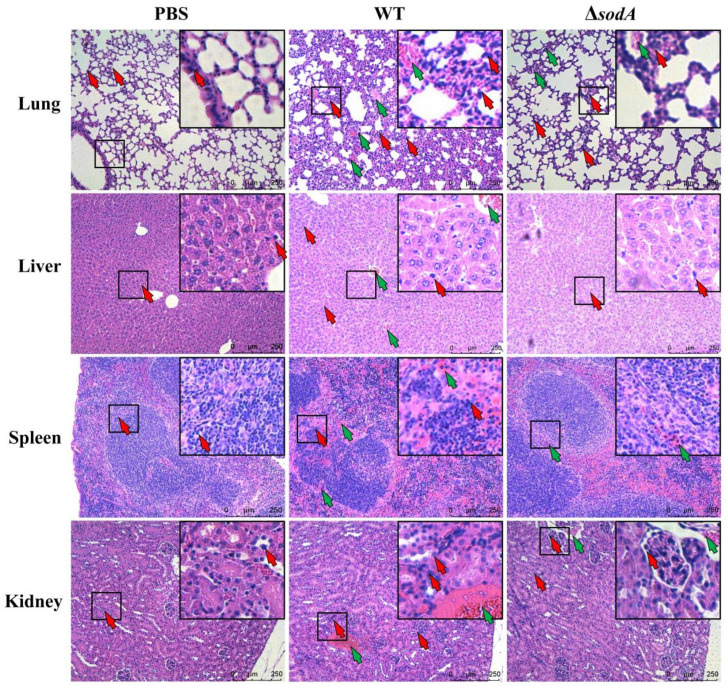
Histological detection of tissue in mice injected with WT and ΔsodA. Tissue samples were collected from mice at 24 h after intraperitoneal injection (*n* = 3); HE staining was used to estimate tissue damage of lung, liver, spleen, and kidney tissue. Green arrows, hemorrhage; red arrows, macrophage; scale bar = 250 μm.

**Figure 3 microorganisms-13-02566-f003:**
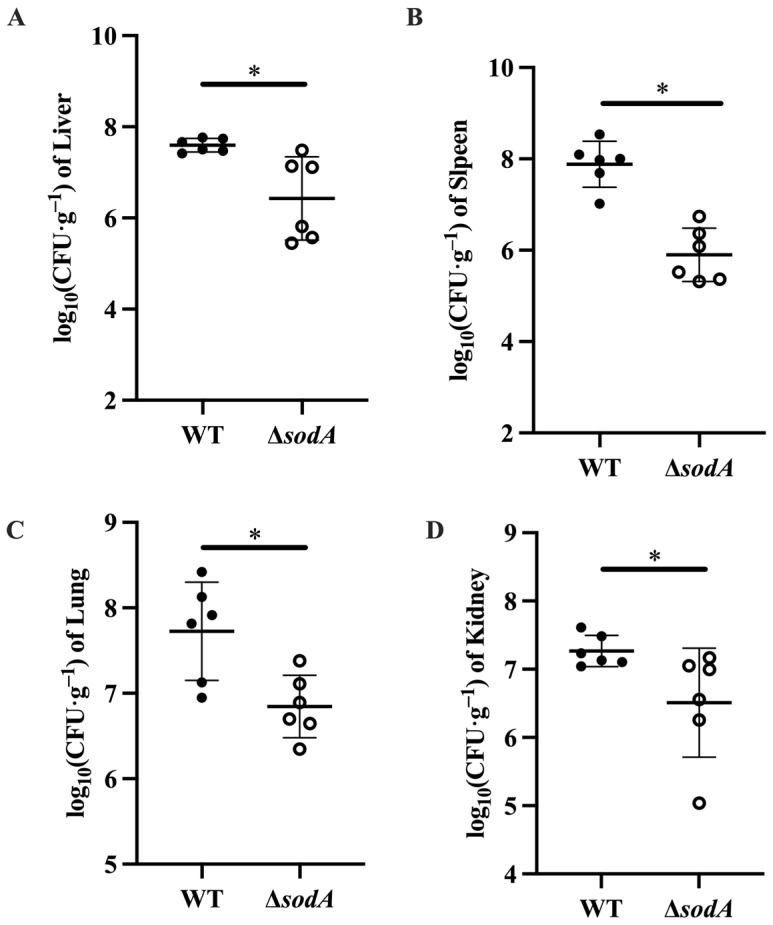
Effect of *sodA* gene deletion on the bacterial colonization of SS2 in mice. Tissue samples were collected from mice at 24 h after intraperitoneal injection, *n* = 6. (**A**) Liver tissue; (**B**) spleen tissue; (**C**) lung tissue; (**D**) kidney tissue. * *p* < 0.05; one-way ANOVA with Tukey’s post hoc test was used; data are shown as mean ± SD.

**Figure 4 microorganisms-13-02566-f004:**
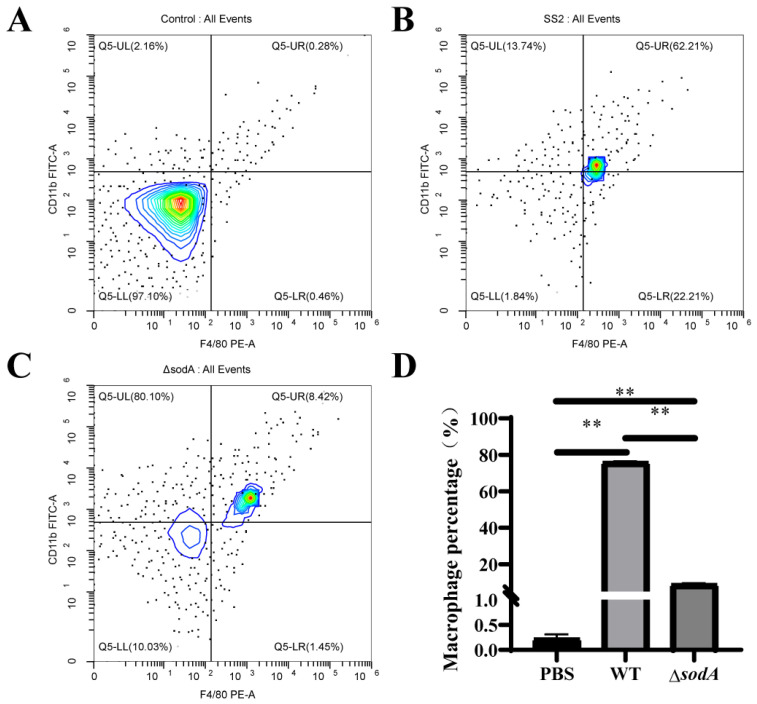
Macrophage population in PLF collected from mice challenged with SS2 and ΔsodA. Macrophages were assessed by CD11b^+^F4/80^+^ and analyzed by flow cytometry. A color scale bar was added to show the cells’ density (blue, green, and red represent low cell density, medium cell density, and high cell density, respectively). (**A**–**C**) PLF samples collected from mice 24 h after intraperitoneal injection with PBS, WT, and Δ*sodA*, respectively. (**D**) The cell numbers in the upper right quadrant (F4/80-positive/CD11b-positive, macrophages) were counted and shown. *n* = 3, ** *p* < 0.01; one-way ANOVA with Tukey’s post hoc test was used; data are shown as mean ± SD.

**Figure 5 microorganisms-13-02566-f005:**
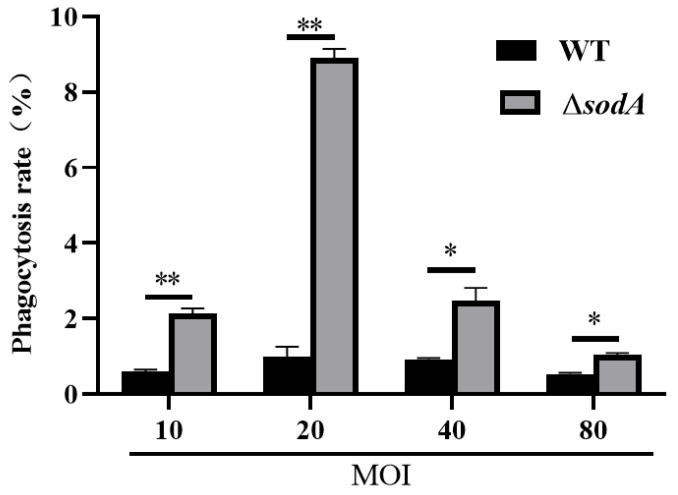
Effect of *sodA* gene deletion on phagocytosis in macrophages of SS2. Phagocytosis of WT and Δ*sodA* was calculated in a macrophage cell line J774A.1 1 h after bacteria were incubated with macrophages (*n* = 3). * *p* < 0.05, ** *p* < 0.01; one-way ANOVA with Tukey’s post hoc test was used; data are shown as mean ± SD.

**Figure 6 microorganisms-13-02566-f006:**
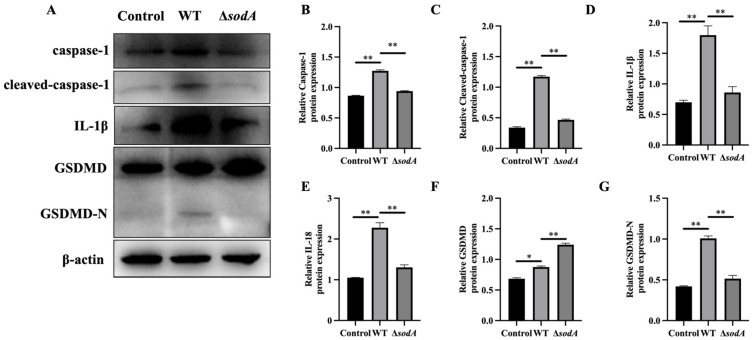
Effect of *sodA* gene deletion on caspase-1/GSDMD pathway activation in macrophages induced by SS2. (**A**) Immunoblotting bands of β-actin, GSDMD, GSDMD-N, caspase-1, cleaved-caspase-1, IL-1β, and IL-18; (**B**–**G**) relative expression of the proteins to β-actin. Macrophages were incubated with WT or Δ*sodA* for 8 h (MOI = 20, *n* = 3). * *p* < 0.05, ** *p* < 0.01; one-way ANOVA with Tukey’s post hoc test was used; data are shown as mean ± SD.

**Figure 7 microorganisms-13-02566-f007:**
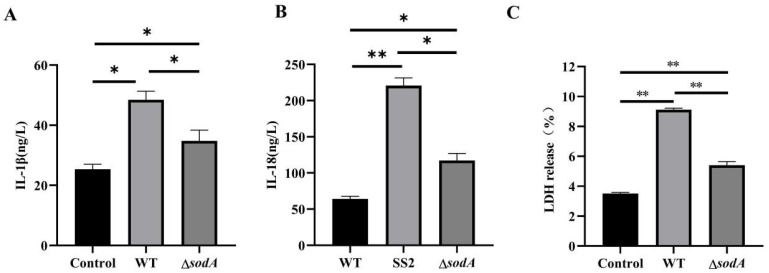
Effect of *sodA* gene deletion on inflammatory cytokine release in macrophages induced with SS2. (**A**–**C**) The contents of IL-1β, IL-18 and LDH in the supernatant. Macrophages were incubated with WT or Δ*sodA* for 8 h (MOI = 20, *n* = 3). * *p* < 0.05, ** *p* < 0.01; one-way ANOVA with Tukey’s post hoc test was used; data are shown as mean ± SD.

## Data Availability

The original contributions presented in this study are included in the article. Further inquiries can be directed to the corresponding author.

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
