# Peer review of "Deletion of the *sodA* Gene Impaired the Pathogenicity of *Streptococcus suis* Serotype 2 to Mice by Inhibiting Caspase-1/GSDMD Pathway Activation in Macrophages"

_microorganisms, 2025, doi:10.3390/microorganisms13112566_

Round 1
Reviewer 1 Report
Comments and Suggestions for Authors
The manuscript, “Deletion of sodA Gene Impaired Pathogenicity of Streptococcus Suis Serotype 2 To Mice by Inhibiting Caspase-1/GSDMD Pathway Activation in Macrophage” by Li et al., describes the effect of sodA gene deletion in Streptococcus Suis serotype 2 on its pathogenesity in mice. The authors found that sodA deletion resulted in less bacterial load, better survival rate, and reduced tissue damage in the organs of infected mice. Additionally, caspase-1/GSDMD pathway was found to be affected in mice infected with sodA deleted SS2. The manuscript is well written with clear images or graphs in each result section. I do not have any specific concern or comment for the authors.
Author Response
Comments 1:
The manuscript, “Deletion of sodA Gene Impaired Pathogenicity of Streptococcus Suis Serotype 2 To Mice by Inhibiting Caspase-1/GSDMD Pathway Activation in Macrophage” by Li et al., describes the effect of sodA gene deletion in Streptococcus Suis serotype 2 on its pathogenesity in mice. The authors found that sodA deletion resulted in less bacterial load, better survival rate, and reduced tissue damage in the organs of infected mice. Additionally, caspase-1/GSDMD pathway was found to be affected in mice infected with sodA deleted SS2. The manuscript is well written with clear images or graphs in each result section. I do not have any specific concern or comment for the authors.
Response 1: We feel great thanks for your professional review work on our manuscript.
Reviewer 2 Report
Comments and Suggestions for Authors
It seems an interesting manuscript with important findings. According to the Introduction, the objective of the study was to investigate the role of sodA gene in pathogenicity of Streptococcus suis serotype 2 (SS2). A deletion mutant for this gene (Δsod) and a wild type strain (HA9801) were compared for virulence to mice. The results indicated that sodA gene was important to the pathogenicity of SS2 accelerating macrophage accumulation and pyroptosis.
In my opinion, the methodology and results sections are well-described. I only recommend further effort to reduce the number of subsections in the Materials and Methods section. Regarding the Results section, I consider it to have a good description of the findings, including informative tables and figures. On the other hand, the Abstract, Introduction, and Discussion sections are poorly written. I suggest the authors improve these three sections:
Abstract: It is difficult to understand the context, objective, and main findings. The writing needs significant improvement to make the message clearer.
Introduction: It is too short and lacks transitions between paragraphs to allow for a good understanding of the topic. I also consider it necessary to include more epidemiological information on SS2 and its importance in the global swine industry.
Discussion: It is too short and contains few comparisons. A paragraph detailing the study's limitations is also missing.
Therefore, I believe the authors need to revise the entire manuscript. A careful review of the English language is also necessary, checking the manuscript text for grammar, style, and syntax.

Author Response
It seems an interesting manuscript with important findings. According to the Introduction, the objective of the study was to investigate the role of sodA gene in pathogenicity of Streptococcus suis serotype 2 (SS2). A deletion mutant for this gene (Δsod) and a wild type strain (HA9801) were compared for virulence to mice. The results indicated that sodA gene was important to the pathogenicity of SS2 accelerating macrophage accumulation and pyroptosis.
In my opinion, the methodology and results sections are well-described. I only recommend further effort to reduce the number of subsections in the Materials and Methods section. Regarding the Results section, I consider it to have a good description of the findings, including informative tables and figures. On the other hand, the Abstract, Introduction, and Discussion sections are poorly written. I suggest the authors improve these three sections:
Response:Thank you for your nice comments, we have reduced the number of subsections in the Materials and Methods section. (Revised on Page 3-4, Line 76-131)
Abstract: It is difficult to understand the context, objective, and main findings. The writing needs significant improvement to make the message clearer.
Response: We feel great thanks for your professional review work on our Abstract, we have revised it according to your comments. (Revised on Page 1, Line 17-32)
Introduction: It is too short and lacks transitions between paragraphs to allow for a good understanding of the topic. I also consider it necessary to include more epidemiological information on SS2 and its importance in the global swine industry.
Response: Thank you for your kind comments, we have revised Introduction according to your advice. (Revised on Page 2-3, Line 38-86)
Discussion: It is too short and contains few comparisons. A paragraph detailing the study's limitations is also missing.
Response: Thank you for your kind comments, we have revised Discussion according to your advice. (Revised on Page 12-13, Line 270-321)
Therefore, I believe the authors need to revise the entire manuscript. A careful review of the English language is also necessary, checking the manuscript text for grammar, style, and syntax.
Response: We sincerely thank the reviewer for careful reading, and have revised the English language in the entire manuscript. (Revised on whole manuscript)
Reviewer 3 Report
Comments and Suggestions for Authors
The whole manuscript needs revision for writing mistakes.
Abstract: revise. Define all abbreviations. Some sentences are unclear, including information about animals, design, etc.
Avoid using "we" and "our".
L37: What is the host?
Introduction: revise; some sentences are not clear. Organize the paragraphs, improve the objectives.
L70-71: relocate
L87: Provide more details about the bacteria preparation.
The methods require more details.
L132: highly significant.
Figure 2: More explanation is required.
Graphs: why n wan not equal in all measurements? Also, some figures have only *P<0.05, others **P<0.01; correct the information under the tables. When you have 3 means, you can use a letter to separate the means.
**P<0.01;
Comments on the Quality of English Language
The whole manuscript needs revision for writing mistakes.
Author Response
The whole manuscript needs revision for writing mistakes.
Response: We sincerely thank the reviewer for careful reading, and have revised the writing mistakes in the entire manuscript. (Revised on Page 12-13, Line 270-321)
Abstract: revise. Define all abbreviations. Some sentences are unclear, including information about animals, design, etc.
Response: Thank you for kind comments, we have defined all abbreviations in Abstract, and added the information you have mentioned. (Revised on Page 1, Line 18-23)
Avoid using "we" and "our".
Response: Thank you for pointing out that, we have revised it in whole manuscript. (Revised on whole manuscript)
L37: What is the host?
Response: Thank you for pointing out that, we have added the information. (Revised on Page 2, Line 39-41)
Introduction: revise; some sentences are not clear. Organize the paragraphs, improve the objectives.
Response: Thank you for your professional comments, we have revised the introduction according to your advice. (Revised on Page 2-3, Line 39-88)
L70-71: relocate
Response: Thanks for your suggestion. we have revised it. (Revised on Page 3, Line 70-76)
L87: Provide more details about the bacteria preparation.The methods require more details.
Response:Thank you for your kind comments. we have revised it. (Revised on Page 3, Line 101-105)
L132: highly significant.
Response:Thank you for pointing out it. We have revised ti. (Revised on Page 5, Line 152)
Figure 2: More explanation is required.
Response:Thank you for kind comments. We have revised ti. (Revised on Page 6, Line 177-188)
Graphs: why n wan not equal in all measurements? Also, some figures have only *P<0.05, others **P<0.01; correct the information under the tables. When you have 3 means, you can use a letter to separate the means.
Response: Thank you for kind comments. In this study, “n=10” was involved in survival of the mice challenged with WT and ΔsodA, “n=6” involved in the bacterial loads experiment (Figure 3), and “n=3” in others. Because the survival rate experiment and bacterial loads are more susceptible to the impact of differences in sample size, we have set a larger sample size.
We have corrected the information in Figure 1, Figure 2, Figure 4.
Round 2
Reviewer 3 Report
Comments and Suggestions for Authors
No further comments